# Folate Receptor Targeted Photodynamic Therapy: A Novel Way to Stimulate Anti-Tumor Immune Response in Intraperitoneal Ovarian Cancer

**DOI:** 10.3390/ijms241411288

**Published:** 2023-07-10

**Authors:** Martha Baydoun, Léa Boidin, Bertrand Leroux, Anne-Sophie Vignion-Dewalle, Alexandre Quilbe, Guillaume Paul Grolez, Henri Azaïs, Céline Frochot, Olivier Moralès, Nadira Delhem

**Affiliations:** 1Univ. Lille, Inserm, CHU Lille, U1189-ONCOTHAI-Assisted Laser Therapy and Immunotherapy for Oncology, 59000 Lille, France; martha.e.b@hotmail.com (M.B.); lea.boidin@inserm.fr (L.B.); bertrand.leroux@inserm.fr (B.L.); anne-sophie.vignon@inserm.fr (A.-S.V.-D.); aquilbe@gmail.com (A.Q.); guillaume.grolez@inserm.fr (G.P.G.); nadira.delhem@inserm.fr (N.D.); 2Department of Gynecological and Breast Surgery and Oncology, Assistance Publique-Hôpitaux de Paris (AP-HP), Pitié-Salpêtrière University Hospital, 75013 Paris, France; henri.azais@aphp.fr; 3Laboratoire des Réactions et Génie des Procédés (LRGP), CNRS-Université de Lorraine, 1 Rue Grandville, 54000 Nancy, France; celine.frochot@univ-lorraine.fr; 4INSERM UMR9020-UMR-S 1277–Canther–Cancer Heterogeneity, Plasticity and Resistance to Therapies, 59000 Lille, France

**Keywords:** photodynamic therapy, ovarian cancer, folate coupled photosensitizer

## Abstract

Photodynamic therapy (PDT) has shown improvements in cancer treatment and in the induction of a proper anti-tumor immune response. However, current photosensitizers (PS) lack tumor specificity, resulting in reduced efficacy and side effects in patients with intraperitoneal ovarian cancer (OC). In order to target peritoneal metastases of OC, which overexpress folate receptor (FRα) in 80% of cases, we proposed a targeted PDT using a PS coupled with folic acid. Herein, we applied this targeted PDT in an in vivo mouse model of peritoneal ovarian carcinomatosis. The efficacy of the treatment was evaluated in mice without and with human peripheral blood mononuclear cell (PBMC) reconstitution. When mice were reconstituted, using a fractionized PDT protocol led to a significantly higher decrease in the tumor growth than that obtained in the non-reconstituted mice (*p* = 0.0469). Simultaneously, an immune response was reflected by an increase in NK cells, and both CD4+ and CD8+ T cells were activated. A promotion in cytokines IFNγ and TNFα and an inhibition in cytokines TGFβ, IL-8, and IL-10 was also noticed. Our work showed that a fractionized FRα-targeted PDT protocol is effective for the treatment of OC and goes beyond local induction of tumor cell death, with the promotion of a subsequent anti-tumor response.

## 1. Introduction

The current challenge in medical oncology is to explore and orchestrate efficient therapeutic options for advanced cancer treatment. Nowadays, it has become clear that such a treatment, in order to be more successful, should destroy the primary tumor while inducing a specific anti-tumor immune response through an immunogenic cell death (ICD) [1]. ICD is a form of regulated cell death sufficient to activate an adaptative immune response in immunocompetent hosts, thus eliciting immunological anti-tumor memory and ensuring long-term efficacy of anti-cancer treatment [2]. In fact, the immune response is a continuous and dynamic mechanism at the same time. This process begins when the immune system can strongly and precisely attack targets based on the principle of antigen-specificity [3]. However, the ability of tumor cells to evade immune detection significantly limits tumor destruction by the immune system [4]. The circumvention of an effective immune response generally combines two large mechanisms: the selection of tumor variants resistant to immune effectors, and the advanced creation of an immune suppressive environment within the tumor. As a consequence, the intent of cancer immunotherapy is to dodge this outcome and to engage the components of the immune system to attack and eliminate tumors by inducing tumor antigen release. This release can happen in several ways: exogenous delivery such as therapeutic vaccine [5], or endogenous delivery by dying tumor cells after the action of certain anti-cancer treatments such as chemotherapy, radiotherapy, oncolytic viruses, and photodynamic therapy [6]. In fact, photodynamic therapy (PDT) based on the use of a non-toxic photosensitizer (PS), a source of light with specific wavelength and oxygen, can induce the generation of reactive oxygen species (ROS) within the tumor cells in which PS is preferentially accumulated.

This oxidative stress subsequently leads to important damages to the tumor vasculature [7] but also to direct tumor cell death by apoptosis [8] or necrosis [9]. These types of cell death can be immunogenic through the release and secretion of inflammatory cytokines and chemokines, leading to the activation of the innate immune system and also the release of damage-associated molecular patterns (DAMPs), thus promoting the maturation of antigen-presenting cells [10]. Consequently, and during a later phase, an adaptive immune memory can develop, hence leading to a systemic immune response playing a role in thwarting tumor recurrence. Some preclinical studies as well as clinical studies underlie the important role of the illumination conditions on both the efficacy of PDT [11] and the activation of immune system [12,13,14]. Briefly, high fluence rates in PDT may induce a rapid depletion in oxygen levels, limiting treatment effectiveness [15]. Another study hypothesized that a light dose may be responsible for an absence of immunity stimulation upon PDT [16]. In turn, other studies suggest that PDT with low fluence rate is capable of inducing an inflammatory response characterized by increases in inflammatory cytokine and neutrophil infiltration into the tumor site. Furthermore, the level of apoptosis was higher with low fluence than high fluence rates [17].

Among all gynecological malignancies, ovarian cancer (OC) is an extremely lethal disease with the second highest mortality rate among gynecological cancer in the world [18]. Epithelial ovarian cancer (EOC) is the most common subtype of OC [19]. Among this subtype, the most familiar and fatal forms are high-grade serous carcinomas (HGSC), usually diagnosed in an advanced stage [20]. Such forms are intrinsically aggressive malignancies, and are hence responsible for most ovarian cancer deaths [21]. In a general manner, the OC is a form of solid tumor that develops, metastasizes, and recurs in the abdominal cavity with a high degree of tumor heterogeneity [22]. Even though a variety of OC subtypes can be diagnosed, they are treated as a single disease [23]. Debulking surgery is imperative for most advanced ovarian patients. Nevertheless, micronodular and tumor colonies spreading within the peritoneal cavity cannot be sufficiently treated by surgery and require expanded chemotherapy [18]. These expanded chemotherapies, including paclitaxel and carboplatine, seem to delay cancer progression without impeding peritoneal recurrences, which concerns 60% of patients [24]. This suggests that microscopic peritoneal metastases may be present and are not eradicated or controlled by the standard care. Therefore, treatment of peritoneal ovarian carcinomatosis needs new lines of therapeutic strategies to be able to treat this microscopic metastasis dissemination by specifically targeting cancer cells. To meet this need, PDT could be used as an adjuvant treatment to surgery and chemotherapy on the condition of using a selective PS for cancer cells, thus avoiding damages to healthy tissue [25,26,27]. Several reports have described that 80% of epithelial ovarian cancers overexpress FRα, while this expression is low in the normal epithelial ovarian tissues [28]. Furthermore, a recent study showed that FRα was well expressed by 75% of patients with microscopic peritoneal metastases (which are the cause of peritoneal recurrence) after complete macroscopic cytoreductive surgery [29]. Moreover, levels of FRα seems to correlate with the grade of ovarian cancer and, therefore, with the overall survival rate [30]. Additionally, FRα is involved in numerous signaling pathways, having a role in the growth and proliferation of cancer cells [31]. Taken together, these elements highlighted FRα as a new promising target in the treatment of ovarian cancer. In this context, and in order to target specifically peritoneal metastases which overexpress FRα, a newly patented PS (patent number WO/2017/016397) coupled with folic acid has been developed by our team. In previous studies, we showed that this PS, the pyropheophorbide a-polyethylene glycol-folic acid (Pyro-PEG-FA), illuminated with 672 nm red light at 1 mW/cm^2^, was capable to specifically target human ovarian tumor cells and induce their death [32,33]. Furthermore, PDT using this PS (Pyro-PEG-FA-PDT) seemed to enhance the activation of the immune system by inducing the release of anti-tumor cytokines while the pro-inflammatory and immunosuppressive cytokines were subdued. In the light of these results, we were interested in testing this Pyro-PEG-FA-PDT in an in vivo mouse model. For this purpose, our first objective was to develop a humanized mouse model of intraperitoneal ovarian cancer by specifically injecting OVCAR3 cells expressing luciferase (OVCAR3-Luc). The expression of luciferase enables us to monitor the development of the tumor in real time, as well as to oversee the regression of the tumor after PDT using different fractionized illumination protocols. The second objective was to investigate the immune response after PDT in our humanized in vivo model. This first task required reconstitution of mice with human peripheral blood mononuclear cells (PBMCs) before treating them with the same PDT protocol previously validated [32]. Seven days and then 30 days after the illumination, we investigated the potential immunostimulating effect of PDT by evaluating the profile of the immune response and the secretion of cytokines.

## 2. Results

### 2.1. In Vitro Results

#### 2.1.1. Evaluation of OVCAR3-Luc Viability upon PDT

The impact of the PDT treatment on luciferase expressing confirmed transfected cells (Appendix A) was validated by mitochondrial metabolism-based assays (Figure 1). Results are presented as the mean of triplicate wells of three independent experiments and normalized compared to the NT condition. Regarding this viability, no significant difference was noted for the Illu group when compared to the NT group. We can notice a slight significant decrease in viability in the PS group compared to the NT group at 1 h and 24 h after PDT (both *p* < 0.01), but this significant decrease was not maintained over time. On the contrary, we can observe a strong and major decrease in viability for the PDT group compared to the NT group. This decrease, observed as soon as 1 h after PDT, was sustained throughout the assay (until 120 h post-PDT) (all *p* < 0.0001). From these results, the OVCAR3-Luc cell line exhibits the same reaction upon PDT than the one reported for the OVCAR3 cell line in a previously published study [34]. This similar reaction highlights that transfection does not alter the sensibility of the cells to PDT.

### 2.2. In Vivo Results

#### 2.2.1. Development of a Humanized In Vivo Mouse Model of Intraperitoneal Ovarian Cancer

OVCAR3-Luc cells were injected intraperitoneally into 12 mice. This injection resulted in tumor growth (Figure 2a,b). The development of cancer was assessed by bioluminescence measures (Figure 2a). An increase in the bioluminescence values was calculated throughout the experience, highlighting a disease progression. Thirty-eight days after tumor cell injection, the animals presented severely distended abdomens full of bloody ascites (Figure 2b(1-2-5-6)) and the dispersed disease within the ascitic fluid formed very distinct aggregates. A few small tumor nodules (<0.5 cm^3^) aggregating in the mesentery were observed as well (Figure 2b(3-4)). We also noticed smaller tumors (<0.25 cm^3^) within the peritoneum; some of them adhered loosely to fat in the pelvic region and intestine.

#### 2.2.2. Before the Reconstitution of Immune System

Evaluation of the PS Uptake within Targeting Tissue

We assessed the accumulation of the PS in ovaries and tumor tissues (Figure 3a) by the quantification of its fluorescence signal (Figure 3b). At 4 h, 6 h, and 24 h upon PS injection, the ovaries and the tumors were subjected to fluorescence imaging. Results are presented as means of three independent experiments (i.e., three different mice) per time point. We can notify a very intense signal intensity at 4 h and 6 h post injection in ovaries and tumor. With time passing, the fluorescence of the PS seems to be accumulating further, resulting in high intensity at 24 h in tumor tissue (*p* < 0.0001) (Figure 3b). The biodistribution of the Pyro-PEG-FA for different time points within liver, pancreas, and kidney can be found in Appendix A. These results not shown warrant further investigation to determine if this could have significant consequence in the case of folate-targeted therapy.

Determination of the Fractionized Illumination Scheme

Two fractionized illumination protocols were evaluated by bioluminescence quantification in order to determine the most effective one (Figure 4). Results are presented as means of two independent experiments (i.e., two different mice) per protocol.

When the protocol consisted of 1 min of illumination followed by 2 min of pause, a stabilization in the bioluminescence was observed from the day 44 (i.e., 7 days after PDT). This stabilization was not observed with the two other groups (Figure 4). In fact, the group, which was subjected to 15 min of illumination followed by 10 min of pause, reflects the same increase in tumor growth as the NT group (Figure 4). From these results, the 1–2 protocol was identified as the most efficient protocol and was applied for the following experiments.

Evaluation of PDT Efficacy

After we validated the fractionized scheme (1 min ON/2 min OFF repeated 45 times), we investigated tumor progression in 12 non-reconstituted mice for the 4 conditions (NT, PS, Illu and PDT) (Figure 5). Results are presented as means of three independent experiments (i.e., three mice) per condition.

Regarding Figure 5, a stabilization in the bioluminescence was observed in the PDT group. Compared to the other groups, the PDT group exhibited a significant change in tumor growth (*p* < 0.001). For the three other groups (PS, Illu, NT), we can notice a continuous increase in the bioluminescence. Moreover, this continuous increase seems similar between these groups. 

Evaluation of the Cytokine Release upon PDT

In order to evaluate cytokine release in the in vivo model of intraperitoneal ovarian cancer, we performed ELISA on the serum of the blood taken from the 12 mice (three mice/condition for the four conditions: NT, Illu, PS, and PDT), 72 h post-treatment (Figure 6).

Compared to the NT group, we detected a significant decrease in the secretion of the TGFβ in the PDT group (*p* < 0.01). On the contrary, the PS and Illu conditions exhibited an increase in this cytokine (Figure 7a).

Regarding the production of the IFN*γ*, the PDT group shows a significant, nearly threefold increase when compared to the NT condition (*p* < 0.01) (Figure 6b). The PS and Illu conditions show no significant difference compared to the NT condition.

#### 2.2.3. After the Reconstitution of Immune System

Evaluation of PDT Efficacy

Twelve mice were reconstituted 38 days after the injection of OVCAR3-Luc cell line. In order to investigate the possible interaction between PDT and immune response, we subjected three of these mice to PDT (R-PDT condition). Three other mice were subjected to illumination only (R-Illu), three other mice received injection of the PS only (R-PS), and the last three mice were not treated (R-NT) (Figure 8). Illumination was performed 7 days after reconstitution (i.e., 44 days after the injection of OVCAR3-Luc).

In the first days following the reconstitution, we can notice a small decrease in the bioluminescence for reconstituted mice (group R-NT, R-PS, R-Illu, R-PDT) contrary to the non-reconstituted mice (NRNT) (Figure 7a). However, this effect seems to disappear over time (from the 10th day post-reconstitution). When reconstituted and subjected to PDT, an important significant decrease in the bioluminescence was observed compared to the NT group (*p* < 0.0001). Furthermore, this decrease seems to be prolongated in time (for 68 days) (Figure 7a).

When compared to non-reconstituted mice subjected to PDT (Figure 7b), the bioluminescence was significantly lower when mice were reconstituted (*p* = 0.0469).

Evaluation of the Immune Cell Response upon PDT

To better understand the immune response elicited by PDT using the targeted FRα photosensitiser, we evaluated it by examining the profiles of selected immune cell populations (Figure 8) and released cytokines (Figure 9) 7 and 30 days after PDT (three mice/group).

Regarding the adaptive immune populations, the percentage of B cells did not rise in the R-PDT groups compared to R-NT group whether at 7 or 30 days after PDT (I = 0.88 and *p* = 0.99, respectively) (Figure 8a).

Concerning the percentage of CD4+ T cells, no significant difference was observed between the reconstituted conditions 7 days after PDT except for R-PS group (*p* = 0.0053). Although at day 30 there was a reduction in the percentage compared with day 7 in all the reconstructed conditions, the percentage was higher for the PDT condition alone, which was the only one to be significant compared with R-NT (*p* < 0.0001) (Figure 8b).

For CD8+ T cells, we noticed an increase in this population for R-illu, R-PS, and R-PDT group, of which the latter is highly significant (*p* = 0.0033; *p* = 0.0017; *p* < 0001, respectively) compared to R-NT. Similarly to CD4+ T cells, even though the percentage on day 30 decreases compared to day 7, it remains the highest for the PDT condition only. Moreover, this percentage is the only one that is significantly higher compared to R-NT group (*p* < 0.0001) (Figure 8c).

Regarding the innate immune populations, we noted a significant increase in the percentage of NK cells 7 days after PDT in the group R-Illu (*p* = 0.0139) and R-PDT (*p* = 0.0003), and 30 days after PDT in the group R-PS and R-PDT compared to R-NT (all *p*-values *p* < 0.0001) (Figure 8a). No significant difference for R-Illu group was noticed 30 days after PDT (*p* = 0.6463).

Regarding myeloid cells, a clear and significant increase can be noticed on day 30 in the R-PDT group only (*p* < 0.0001).

Finally, concerning monocytes, no differences between all of the conditions were noticed.

Evaluation of the Cytokines’ Release upon PDT

Regarding the cytokine release, several cytokines were tested and reported (Figure 9 and Appendix A) (three mice/group).

A critical decrease in TGFß was detected 7 days after PDT compared to NRNT (Figure 9a). Thirty days after PDT, the expression of TGFß seemed to be decreasing in all conditions; however, it remained the lowest when reconstituted mice were treated with PDT. By the same token, the expression of the cytokines IL-8 and IL-10 (Figure 9b,c) decreased significantly 30 days after PDT when compared to NRNT condition.

On the contrary, the cytokine IFN*γ* (Figure 9d) was significantly highly expressed 7 days only upon PDT (*p* < 0.001). Moreover, the expression of IFN*γ* did not engage any changes until 30 days upon PDT and remained significantly higher 30 days after PDT (*p* < 0.001).

The expression of TNFα was also significantly boosted 7 days after PDT (*p* < 0.001) (Figure 9e). This boost seems to be persistent until 30 days after PDT when compared to the NRNT conditions (*p* < 0.001).

All the other investigated cytokines were not detectable (Appendix A).

## 3. Discussion

The management of recurrent ovarian cancer (OC) has improved in recent years. The addition of Poly(ADP-ribose) polymerase inhibitors (PARP) and anti-angiogenic agents to conventional treatments has significantly increased the progression free survival (PFS) of these patients [35,36,37]. However, despite major advances in the management of this cancer, 70% of patients in remission will relapse within 18 months [38]. This suggests that microscopic peritoneal metastases may be not eradicated or controlled by the standard care [18]. Amate et al. showed in a retrospective study that the peritoneum was the preferred site of recurrence, since 75% of advanced stage relapses were in this location [39]. Thus, the existence of this residual peritoneal metastasis represents the major problem of this pathology. It is necessary to control this problem by developing new therapeutic strategies with intraperitoneal diffusion. To meet this need, photodynamic therapy could be a good alternative.

An effective PDT is a treatment capable of delivering light in the PS-targeted tissue with the adequate wavelength and dosimetry in order to induce cancer cell death. Constitutionally, the photodynamic process depends on the photosensitizer molecule itself. In this article, we have used a Pyro-PEG-FA-PDT that has already shown its effectiveness in an in vitro model [34]. Using an illumination device developed in the laboratory [40] and dedicated to in vitro PDT studies (1 mW/cm^2^, 3.6 J/cm^2^), we were able to provide proof of concept of an effective PDT based on this PS coupled with folic acid on an in vitro ovarian cancer model. Such a PDT was, in fact, demonstrated to induce almost 100% of cell death on ovarian cancer cells [34].

In the present paper, we established a stable cell line expressing luciferase (OVCAR3-Luc) and we tested the efficacy of this PS on this new cell line using the same illumination device. As expected, upon illumination and in the presence of Pyro-PEG-FA, OVCAR3-Luc exhibited important cell death 24 h after illumination. Compared to the OVCAR3 cell line used in our previous study [34], no changes were distinguished upon PDT, suggesting that the transfection of the luciferase does not disturb the sensibility of these cells to PDT.

After validation of our cell line, we injected 1 × 10^6^ OVCAR33-Luc cells into the intraperitoneal cavity of SCID mice. Using the bioluminescence properties of luciferase, we were able to monitor the development of the intraperitoneal OC. The increase in bioluminescence emphasized the development of the cancer inside the peritoneal cavity of the mice. The establishment of a mouse ovarian cancer has already been reported [41]. Compared to this other article, we have chosen to inject the mice with a smaller quantity of cells. The purpose was to have a slow but effective tumor development, allowing us to follow it meticulously in time and to eventually assess the development or regression of cancer growth upon PDT. As described in the result, upon 38 days of OVCAR3-Luc cell injection, we had a constant intraperitoneal tumor growth. Though commodious and ubiquitous, the metastases in SCID mice were enclosed in pelvic and abdominal cavities, as this is the case for the common metastatic pattern in women diagnosed with ovarian cancer Stage III [42].

With regard to the uptake of Pyro-PEG-FA, it is necessary that the uptake of the therapeutic concentration of PS is specifically achieved by the target cells. High selectivity of the PS for the neoplastic tissues is then an important aspect of the success of PDT [43]. As shown in our result, upon 24 h of PS injection, the fluorescence is well concentrated in the tumor tissues and the ovaries, thus confirming a good targeting of the Pyro-PEG-FA.

After validated Pyro-PEG-FA accumulation within targeted tissue, upon 24 h of Pyro-PEG-FA injection, we evaluated the Pyro-PEG-FA-PDT effect based on two fractionized protocols: 1 min of illumination followed by 2 min of pause repeated 45 times, and 15 min of illumination followed by 10 min of pause repeated three times. The overall duration of illumination was 45 min. For this purpose, we used an illumination device dedicated to in vivo studies of mice. This device, which delivers red light at 10 mW/cm^2^ (light dose for 45 min of illumination: 27 J/cm^2^), has already been used in a preclinical study on PDT for human pancreatic adenocarcinoma [32]. The first protocol (1 min ON/2 min OFF) induced a higher continuous stabilization in the bioluminescence. This observation is supported by the need of oxygen to achieve a PDT effect. Indeed, despite major advances in the use of PDT as an anti-cancer therapy, one of its limiting factors remains its potential ineffectiveness in the event of a hypoxic tumor or when hypoxia occurs due to the consumption of molecular oxygen during PDT photochemical reactions [44]. As a matter of fact, the level of tissue oxygen is stirred by the illumination protocols applied. It has been demonstrated several times that a fractionated protocol makes it possible to maintain a high level of molecular oxygen, thus improving PDT efficiency compared to a continuous illumination protocol [45,46]. That is why we directly have tested two fractionized protocols. The irradiation is also recognized as a key factor that determines the efficacy of PDT. Some studies have shown a significant decrease in oxygen partial pressure (pO_2_) with high irradiance (>90 mW/cm^2^) contrary to lower irradiance (5 and 30 mW/cm^2^), which allowed the maintenance of pO_2_ at levels comparable to those measured before illumination [47]. Another study showed that low irradiance (20 mW/cm^2^) was more efficient than high irradiance (50 mW/cm^2^) in normal mouse skin [48]. Accordingly, the irradiance that we used (10 mW/cm^2^) was expected to significantly decrease the bioluminescence when PDT was applied to mice, and this was demonstrated by our results. Compared to the other controls, only the PDT condition showed a noteworthy decline in bioluminescence.

Moreover, and when mice were reconstituted with human PBMCs, a significant decrease in the bioluminescence was also described. More importantly, when compared to the non-reconstituted mice, there was a higher decrease of the tumor size. Our results support the hypothesis suggesting that PDT can enhance the activity of immune cells in favor of an anti-tumor response [34]. The anti-tumor effects of PDT can be outlined in three actions: tumor cell photodamage via ROS, tumor-associated vascular damage, and initiation of immune response [49]. Regarding the latter, PDT can favor acute tissue-based inflammatory response, leukocyte infiltration on the tumor site, and production of proinflammatory cytokines [50]. That is why we investigated immune population in our model. Concerning innate immune cells as NK cells and myeloid cells, we traced an increase 30 days after PDT in the R-PDT group. Lastly, we noticed an increase in the CD4+ T and CD8+ T lymphocytes 30 days after PDT in the PDT group only, highlighting the establishment of an adaptive anti-tumor immune response. Finally, we have not seen any change in the percentage of LB cells, thus suggesting that the effect of PDT does not involve the activation of the humoral response.

Therefore, it seems that the effective immune system plays an important role in the effectiveness of PDT as the decrease in the tumor growth was higher in the reconstituted group than the non-reconstituted group. Furthermore, another study has also explored the role of the host immune system in contributing to tumor regression after PDT and showed that CD8+ T was required to prevent tumor regrowth and NK cells contributed to this response [51].

Concerning cytokine secretion, even when the mice were not reconstituted, we were able to portray a decrease in the fold of expression of TGFß and an increase in IFN*γ*. Arguably, major benefits might be accomplished with this immunostimulant capacity of PDT. Regarding this tissue-based inflammation, a lot of articles described that PDT could enhance a successful passage from innate to adaptive anti-tumor immunity, hence converting the immunosuppressive tumor microenvironment into a more favorable anti-tumor one [52]. We have previously shown that in an in vitro model of ovarian cancer, both cytokines were regulated upon PDT [34]. Even more interestingly, in the present paper, when mice were reconstituted, several changes in the cytokine expression were reported. In particular, the cytokine TGFß decreased more significantly 7 days upon PDT in the PDT condition compared to the other conditions. Regarding the cytokines IL-8 and IL-10, their expression was the lowest 30 days upon PDT in the PDT group. These cytokines are known to have an unfavorable role in ovarian cancer. Indeed, their high serum value has been closely linked to the stage and prognosis of ovarian cancer and their concentration in the peritoneal fluid has been associated with cancer progression and appears to be a factor of poor prognosis [53,54]. Furthermore, IL-8 has been described as promoting migration, invasion, and epithelial-mesenchymal transition of ovarian cancer cells [55], while IL-10 tends to reduce activation of dendritic cells, thus impacting the activation of T cells [56]. Thus, the downregulation of these cytokines by PDT that we observed in the R-PDT group could reduce the aggressiveness phenotype of the tumor.

Lastly, the cytokine IFN*γ* and TNFα were highly expressed 7 days upon PDT. Moreover, this expression persists until 30 days post-PDT. These results suggest that PDT is capable of promoting the secretion of these cytokines participating in an immunoactivating effect. In fact, these cytokines were described as a major player in immune activation after PDT turned it towards the main antitumor immune response, the TH1 pathway [1]. Indeed, IFN*γ* is a cytokine capable of preventing tumor development through its ability to inhibit angiogenesis, to orchestrate the attraction and maturation of CD4+ and CD8+ lymphocytes, to promote the activation of numerous cell types such as monocytes and NK cells, and to regulate the production of immunoglobulins by B lymphocytes [57]. As for the pro-inflammatory cytokine TNFα, it is secreted in the physiological state by macrophages, stromal cells and lymphocytes. Moreover, it has been shown that PDT was able to induce an acute inflammation characterized by the secretion of this cytokine, which can promote the activation of T lymphocytes and, thus, an adaptive anti-tumor immune response as previously described [58]. Therefore, it would be interesting to question the mechanism of an immunogenic cell death (ICD) induced upon PDT. In fact, ICD aids the release of damage-associated molecular patterns (DAMPs) from dying tumor cells, resulting in the activation of the tumor-specific immune responses that we observed [59].

Taken as a whole, our study shows that our Pyro-PEG-FA-PDT is capable not only of inducing tumor cell death but also of activating immune response in favor of an anti-tumor response. In our mouse intraperitoneal ovarian cancer model, the PDT effect on immune response was prolonged in time, hence insuring a continuous tumor cell death aftermath.

## 4. Materials and Methods

### 4.1. In Vitro

#### 4.1.1. Cell Culture

The ovarian tumor cell line (OVCAR3) was ordered from the American Type Culture Collection (ATCC). OVCAR3 were cultured in RPMI-1640 medium supplemented with 10% heat-inactivated fetal calf serum (Gibco, Thermo Fisher Scientific, Waltham, MA, USA) and 1% penicillin (Gibco, Thermo Fisher Scientific, Waltham, MA, USA). Cells were maintained in an incubator at 37 °C, 5% CO_2_, and 95% humidity.

#### 4.1.2. Luciferase Transfection

Stable luciferase (luc)-transfected clones of the ovarian cancer cell line (OVCAR3) previously described (cf., Section 2.1.1) were generated by transfection with pGL3 vector containing the luc gene. (Promega, Leiden, The Netherlands) (Appendix A). Briefly, one day before transfection, 0.5–2 × 10^5^ cells were cultured in 500 μL of growth medium without antibiotics. DNA was diluted in 50 μL of RPMI-1640 medium without serum. Lipofectamine 2000 was diluted (DNA (μg): Lipofectamine 2000 (μL) ratios from 1:0.5 to 1:5 in 50 μL of Opti-MEM I medium and incubated for 5 min. Upon incubation, both of the solutions were mixed incubated for 20 min at room temperature. The mixture was then added to 100 μL of complexes in each well containing cells and medium. The OVCAR3-Luc cell line was then incubated at 37 °C in a CO_2_ incubator for 48 h. After transfection, the cells were cultured in RPMI-1640 medium supplemented with 10% heat-inactivated fetal calf serum (Gibco, Thermo Fisher Scientific, Waltham, MA, USA) and 1% penicillin (Gibco, Thermo Fisher Scientific, Waltham, MA, USA) with passages with Neomycin (Sigma Aldrich, Saint Louis, MO, USA) at 10 μg/mL every one month. Cells were maintained in an incubator at 37 °C, 5% CO_2_ and 95% humidity.

#### 4.1.3. Luciferase Assay

In order to evaluate the success of the transfection, a luciferase assay was performed. OVCAR3-Luc and OVCAR3 cells were lysed (at 24 h post-transfection for OVCAR3-Luc), and luciferase gene expression was quantified using a commercial kit (Promega, Cergy Pontoise, France) and a luminometer (Centro LB960, Berthold Technologies, Bad Wildbad, Germany). Results are presented as means of three independent experiments, expressed as relative light units (RLUs) integrated over 10 s per milligram of cell protein lysate (RLU/mg protein).

#### 4.1.4. Photosensitizer

The present study used a new PS patented by our lab (patent number WO/2017/016397). Its chemical structure is shown in the following Figure 10 and is based on folic acid conjugated to pyropheophorbide-a via a polyethylene glycol type spacer (Pyro-PEG-FA). As for the synthesis, it has already been published and described by our lab in the patent quoted above.

#### 4.1.5. Photodynamic Therapy In Vitro

For in vitro assays, we repeated the same protocol described by Baydoun et al. [34] using the illumination device developed in our laboratory as described elsewhere [40] Four groups of OVCAR3-Luc cancer cells were used: untreated cells (NT), cells treated with PS without illumination (PS), cells treated without PS with illumination (Illu), and cells treated with PS and subjected to illumination (PDT). Briefly, 2000 OVCAR3-Luc cells were seeded per well in a 96-well, white-walled, and clear-bottomed plate (Corning, Somerville, MA, USA). After 24 h, cancer cells were treated (PS; PDT) or not (NT; Illu) with 9 µM of PS. Another 24 h later, the medium containing the PS or not was changed and replaced with the normal medium of the cell type after two washing steps with PBS (Gibco, Thermo Fisher Scientific, Waltham, MA, USA). A homogeneous illumination (1 mW/cm^2^) was then performed for the groups Illu and PDT during 1 h (3.6 J/cm^2^), with a specific 672 nm laser-based device developed by our OncoThAI research unit [40]. This device includes a heating module, enabling us to maintain the cells at a temperature of 37 °C. All the experiments were performed in dark conditions. Results are presented as means of three independent experiments expressed in % of viability and normalized compared to NT condition.

#### 4.1.6. Viability Assays

OVCAR3-Luc ovarian cancer cell line viability after PDT was assessed by a viability assay based on mitochondrial metabolism (CellTiterGlo^®^, Promega, Madison, WI, USA). Briefly, 2000 cancer cells were seeded per well in a 96-well, white-walled, and clear-bottomed plate (Corning, Amsterdam, The Netherlands) per time point and then subjected to above-described PDT treatments. Then 1 h, 24 h, 48 h, and 72 h post-treatment, 100 µL/well of the Celltiter-Glo mix was added at room temperature for 10 min and protected from light according to manufacturer’s instructions. The bioluminescence was then read using a luminometer (Centro LB960, Berthold Technologies, Bad Wildbad, Germany) under the Microwin software v4.41.

### 4.2. In Vivo

#### 4.2.1. Animals

Animal experiments were performed according to the rules for care and use of experimental animals, conducted in accordance with the local ethics Committee of the Institut Pasteur de Lille and performed with required permission of the National governing ethical board (approval number 2019041015585930). Female SCID mice (Charles Rivers Laboratories, Saint Germain Nuelles, France) were used for all the experiments. All female mice were aged 6 to 8 weeks and kept in a pressurized and ventilated cage (6 mice/cage) with a regular mouse diet of 10% animal fat. For the experiment under anesthesia, mice were anesthetized using inhalation of isoflurane (5% for anesthesia induction and 2% thereafter).

#### 4.2.2. Development of a Humanized Mouse Mode of Intraperitoneal Ovarian Cancer

A splenectomy was performed on anesthetized SCID mice before cancer cell injection. Seven days after the splenectomy, the mice were inoculated intraperitoneally with 1 × 10^6^ OVCAR-3 Luc cells at four points. Due to the luciferase’s properties of bioluminescence, tumor growth was supervised by using intraperitoneal injection of 100 μL of D-luciferin (30 mg/mL, Perkin Elmer, Waltham, MA, USA) over an IVIS LUMINA XR reader (Caliper Life Sciences, Hopkinton, MA, USA), and analyzed under Living Image 4.1 software (Caliper Life Sciences, Hopkinton, MA, USA). Results were obtained after spectral unmixing according to the manufacturer’s instructions and expressed in bioluminescence.

#### 4.2.3. Before the Reconstitution of Immune System (Non-Reconstituted (NR) Mice)

PS Uptake

Thirty-eight days after the mice were inoculated with the OVCAR3-Luc cell line, 100 μL of PS solution at 1 mg/100 mL dissolved in PBS was injected intraperitoneally in order to evaluate the accumulation of the PS in different organs. This evaluation relies on the fluorescence properties of the PS under 412 nm blue light. At 4 h, 6 h, and 24 h after the injection, mice were euthanized. Ovaries and tumors were removed (3 mice/time point) and subjected to fluorescence imaging by Multiphoton microscope Leica SP8 (Leica, Wetzlar, Germany). Images were subsequently analyzed under ImageJ 1.53a software and a semi-quantification of the signal was performed. Results are expressed as fluorescence intensity and compared to NC (NC: negative control = mice that did not receive the PS).

Determination of the Fractionized Illumination Scheme

Twenty-four hours after the inoculation of the PS solution, 2 protocols of fractionized illumination were assessed: 15 min of illumination followed by 10 min of pause repeated 3 times (group 15–10), or 1 min of illumination followed by 2 min of pause repeated 45 times (group 1–2). This second protocol was derived from a study previously published by Quilbe et al., [32]. Whatever the protocol, the overall duration of illumination was 45 min. Both protocols involved an irradiance of 10 mW/cm^2^ and therefore delivered the same light dose of 27 J/cm^2^. A 672 nm homemade laser-based device, specifically designed for in vivo PDT of mice, was used for the illumination. The experiment was performed in comparison with a third group of non-treated mice (group NT). For 15–10 and 1–2 groups, 100 μL of PS solution at 1 mg/100 mL dissolved in PBS was injected intraperitoneally into the mice, 24 h prior to illumination. Post PDT, mice were frequently subjected to bioluminescence imaging in order to follow the time effect of PDT on the tumor growth. Protocol efficacy was examined for 54 days and gauged by bioluminescence quantification. Images were subsequently analyzed under the Living Image 4.1 software (CaliperLife Sciences, Hopkinton, MA, USA) and results were obtained after spectral unmixing according to the manufacturer’s instructions. From these results, the most efficient protocol was identified and applied for the following experiments.

Photodynamic Therapy Protocol

Twelve mice were divided into four groups: untreated mice (NT), mice treated with PS but without illumination (PS), mice treated without PS but with illumination (Illu), and mice treated with PS and subjected to illumination (PDT). For PS and PDT groups, 100 μL of PS solution at 1 mg/100 mL dissolved in PBS was injected intraperitoneally into the mice 24 h prior to illumination. For Illu and PDT groups, mice were subjected to the fractionated illumination protocol determined in the previous experiment (cf., Section 2.2.2). The effect of PDT on the tumor growth was assessed through frequent bioluminescence imaging. Images were subsequently analyzed, and results obtained as previously described. Results were expressed in log(10)bioluminescence.

ELISA

Cytokine detection was carried out 72 h after PDT on mice serum treated or not with PDT as previously described. Transforming growth factor (TGF)-β1, and interferon (IFN)-γ secretion were determined by the Sandwich ELISA (Enzyme-Linked ImmunoSorbent Assay) method. Briefly, purified primary antibodies were coated overnight at 4 °C in flat-bottomed 96-well maxisorp plates (NUNC, Thermo Fisher Scientific, Waltham, MA, USA) before incubation with samples. The corresponding biotinylated antibodies were added for protein detection after several steps of non-specific site blocking, sample deposition (overnight at 4 °C), and adequate washing (PBS-Tween 0.05%). The reaction was amplified with Streptavidine-peroxydase (Interchim, Montluçon France). Cytokine concentrations were finally highlighted with the addition of OPD (10 mg/mL, Sigma-Aldrich, St. Louis, MO, USA). After color development, the plates were read using a Multiskan spectrophotometer at 492 nm powered by Ascent™ Software v2.06 (Multiskan RC Thermo Labsystems, Thermo Fisher Scientific, Waltham, MA, USA). The purified and biotinylated antibodies used were as follows: rat anti-human TGFβ1 and mouse anti-IFN-γ (all from BD PharmingenTM, San Jose, CA, USA). Five groups are presented: before PDT mice, non-treated mice (NT), mice treated with PS but without illumination (PS), mice treated without PS but with illumination (Illu), and mice treated with PS and subjected to illumination (PDT). Results are expressed as normalized values to the NT group pg/mL as the mean of triplicate wells after subtracting background values.

#### 4.2.4. After the Reconstitution of Immune System (Reconstituted (R) Mice)

Reconstitution of Immune System

In order to evaluate the impact of PDT on the immune system, we reconstituted the immune systems of mice with the injection of human peripheral blood mononuclear cells (PBMCs). To this end, human blood samples were collected from healthy human adult donors after obtaining informed consent, in accordance with the approval of the Institutional Review Board at the Biology Institute of Lille (DC-2013-1919). PBMCs were isolated from peripheral blood samples by density gradient centrifugation using lymphocyte separation medium (Eurobio, Les Ullis, France) and leucosep 50 mL tubes (Greiner Bio One, Courtaboeuf, France). Obtained purity was over 95%. Mice were then reconstituted (group R) by intraperitoneal injection of 40 × 10^6^ PBMCs isolated as previously described. Reconstituted mice (group R) were compared to a non-reconstituted group of mice (group NR).

Photodynamic Therapy Protocol

Fifteen mice were divided into 5 groups: reconstituted–not treated mice (R-NT), reconstituted mice treated with PS but without illumination (R-PS), reconstituted mice treated without PS but with illumination (R-Illu), reconstituted mice treated with PS and subjected to illumination (R-PDT), and finally non-reconstituted–not treated mice (NRNT). For R-PS and R-PDT groups, 100 μL of PS solution at 1 mg/100 mL dissolved in PBS was injected intraperitoneally into the mice, 24 h prior to illumination. For R-Illu and R-PDT groups, mice were subjected to the fractionated illumination protocol determined in the previous experiment (cf., Section 2.2.2). The effect of PDT on the tumor growth was assessed through frequent bioluminescence imaging. Images were subsequently analyzed, and results were obtained as previously described. Protocol efficacy was examined for 68 days and gauged by bioluminescence quantification expressed in log(10) bioluminescence.

Flow Cytometric Analysis

Seven days and 30 days after the treatment, cell immunophenotype was analyzed by flow cytometry using Attune NxT (ThermoFisher, Waltham, MA, USA). After PBMC isolation from mouse blood samples, cells were washed with phosphate-buffered saline (PBS) (GIBCO-Life technologies) and labeled with fluochrome-conjugated mAbs (1:10). After their harvest, 5.10^5^ cells were taken up in a volume of 100 μL of PBS^−/−^ (GIBCO-Life technologies, GB) and the fragment crystallizable receptors (FCR) were blocked with the FCR blocking reagent (Miltenyi Biotec, Bergisch Gladbach, Germany) for 15 min at 4 °C. A viability control was carried out by adding FVS 700 (BD life Sciences) for 10 min at room temperature (RT). Then, the cells were labeled for 30 min at 4 °C in the dark with fluochrome-conjugated mAbs: CD14-PE, CD11c-APC, CD3–PECy7, CD4-Vioblue, CD8-VioGreen, CD19-FITC, CD335-APC (Miltenyi Biotech, Bergisch Gladbach, Germany). For each assay, an unmarked control and the appropriate isotypic control mAbs were used for positive signal setting. Median fluorescence intensity (MFI) data were analyzed with FlowJo Software version 2.9.0 (Ashland, OR, USA). Results were expressed as normalized values compared with the R-NT condition.

High Sensitivity Human Cytokine Magnetic Bead Kit

Plates were prepared as per the manufacturer’s instructions. Briefly, each plate was blocked with wash buffer for 10 min before use. The mixed beads were dispensed into each well and washed twice. The standard curve was generated by reconstituting the high sensitivity human cytokine standard, per the manufacturer’s protocol, with serial 1:5 dilutions for a working concentration range of 0.13–400 pg/mL. The mouse serum samples were collected seven days and 30 days after the illumination and standards were incubated with the mixed beads overnight at 4 °C while shaking. The beads were washed and then incubated with a detection antibody at room temperature for 1 h and with streptavidin for an additional 30 min. The beads were washed twice, resuspended in Luminex MagPix^®^ drive fluid, and the plate was subsequently analyzed on the Luminex MagPix^®^ plate reader. Results were expressed in normalized bioluminescence. The MFI was then compared and adjusted for the background to the standard curve to calculate the cytokine concentration in pg/mL. Interleukine(IL)-2, 4, 6, 8, 10, 17, 22 and TGF-b, IFNg, TNFa concentrations were investigated. Results are presented as normalized values compared to non-reconstituted not treated (NRNT) condition. Each standard curve was then individually analyzed for outliers and adjusted as necessary to achieve linearity (R^2^ ≥ 0.8).

### 4.3. Statistical Analysis

All results are expressed as the means and standard deviations of triplicates of at least three independent experiments. All data were analyzed using the statistical package GraphPad Prism for Windows 3.0.1 (GraphPad, San Diego, CA, USA). The normalities of the distributions were assessed using the Shapiro–Wilk test. Differences in the various effects of PDT (Pyro-PEG-FA uptake, sensitivity to PDT, cytokine release, and impact on immune response upon PDT) were assessed using ANOVA and Tukey’s multiple comparisons tests. One-way ANOVA with the treatment group as the factor was used for cytokine release in non-reconstituted mice, whereas two-way ANOVA were carried out for the other analyses. Treatment group and time were included as cofactors in the two-way ANOVA, except for the analysis of Pyro-PEG-FA uptake, in which time and targeting tissue were considered as covariates. Due to the non-normality of data, the overall tumor progression throughout the experiment between non-reconstituted mice and reconstituted mice was assessed using the Mann–Whitney test. All the significance levels were set to 0.05. All quoted *p*-values are two-sided.

## 5. Conclusions

In the present study, we evaluated the effect of PDT using a new targeted photosensitizer in an in vivo intraperitoneal ovarian cancer model treatment. We showed that this PS upon illumination could induce decrease in the tumor volume. Furthermore, when mice were reconstituted with human PBMCs, PDT using this new Pyro-PEG-FA seemed to accentuate the bioluminescence decrease and to ensure a prolonged effect in time. More importantly, PDT could activate an immune response by inducing the secretion of anti-tumor cytokines and increasing anti-tumor lymphocyte populations while inhibiting the pro-inflammatory and pro-tumor cytokines.

The findings presented in this study suggest that for intraperitoneal ovarian cancer, PDT treatment regimens using a folate-receptor targeted PS can flesh out a systemic anti-tumor immune response. In fact, most of the therapies currently used to treat this cancer have uncooperative effects on the host’s immune response. The readiness of a tumor adjuvant treatment adequate for eliminating the tumor and, in concert, stimulating anti-tumor immunity would a major advantage.

## Figures and Tables

**Figure 1 ijms-24-11288-f001:**
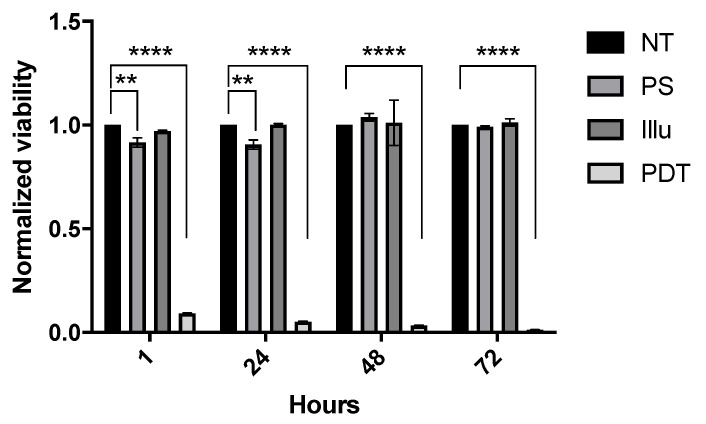
In vitro sensibility of OVCAR3-Luc cell line to PS-PDT over time: normalized viability for OVCAR3-Luc cell line at 1 h, 24 h, 48 h, and 72 h post-illumination. Cells are treated with 9 µM of PS upon 60 min of illumination (1 mW/cm^2^, 3.7 J/cm^2^). NT: non-treated, PS: photosensitizer only, Illu: illumination only, PDT: illumination in the presence of PS. Results are presented as means of three independent experiments expressed in % of viability and normalized compared to NT condition. Two-way ANOVA statistical test was performed, all quoted *p*-values are two-sided, with *p* ≤ 0.01 (**), and *p* ≤ 0.0001 (****) being considered statistically significant for the first and highly significant for the others. *n* = 3.

**Figure 2 ijms-24-11288-f002:**
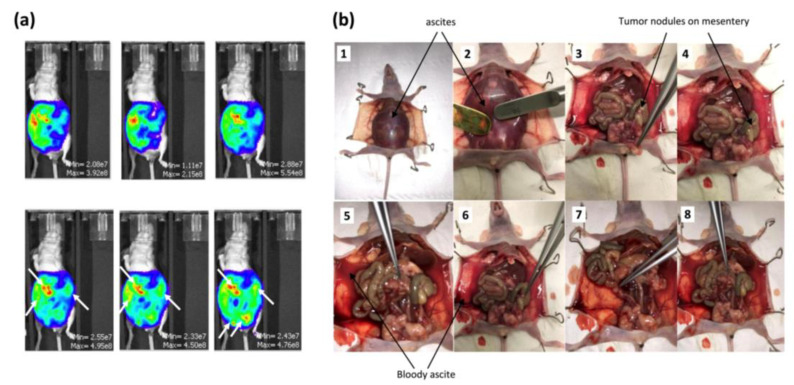
Development of the in vivo humanized SCID model of intraperitoneal ovarian cancer: (**a**) evaluation of the tumor growth by bioluminescence after injection of 100 µL of D-luciferin over an IVIS LUMINA XR reader 38 days after tumor cell injection; (**b**) OVCAR3-Luc cells injected into SCID mice resulted in tumor growth 38 days after cell injection.

**Figure 3 ijms-24-11288-f003:**
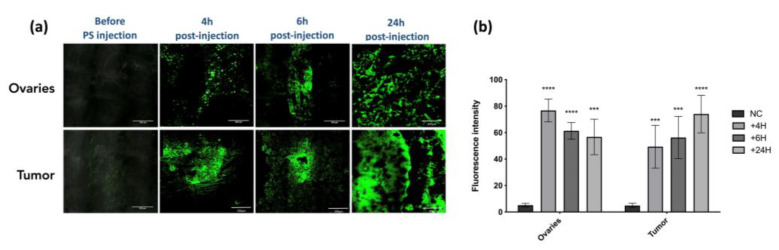
PS uptake: (**a**) evaluation of the uptake of the PS within ovaries and tumor in an in vivo humanized SCID mice model of intraperitoneal ovarian cancer (scale bar = 200 µm); (**b**) semi fluorescence quantification of the signal by ImageJ software (version 2.9.0) (NC: negative control). Results are expressed in fluorescence intensity 4 h, 6 h, and 24 h after PS injection and presented as means of three independent experiments. Two-way ANOVA statistical test was performed, all quoted *p*-values are two-sided, with *p* ≤ 0.001 (***) and *p* ≤ 0.0001 (****) being considered statistically highly significant. *n* = 3.

**Figure 4 ijms-24-11288-f004:**
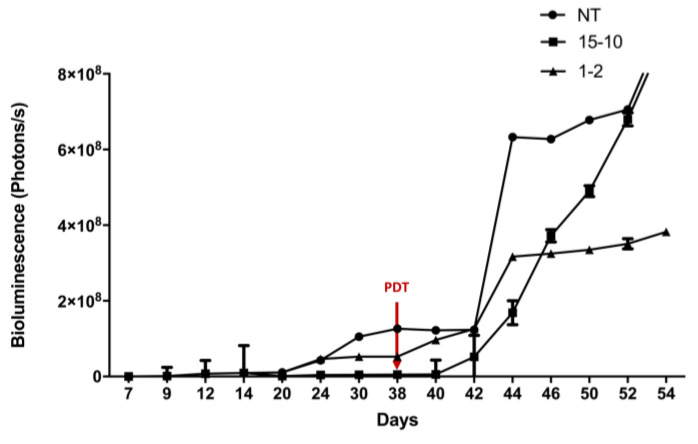
Determination of the most effective fractionized illumination protocol for PDT in humanized non-reconstituted SCID mice model of intraperitoneal ovarian cancer: mice were subjected to two different fractionized illumination protocols: 1 min illumination followed by 2 min of pause (1–2) vs. 15 min illumination followed by 10 min of pause (15–10), both delivering 27 J/cm^2^ red light at 10 mW/cm^2^. Comparison with NT: non treated group was performed. Arrow: day of illumination. Results are presented as means of two independent experiments per protocol. *n* = 2.

**Figure 5 ijms-24-11288-f005:**
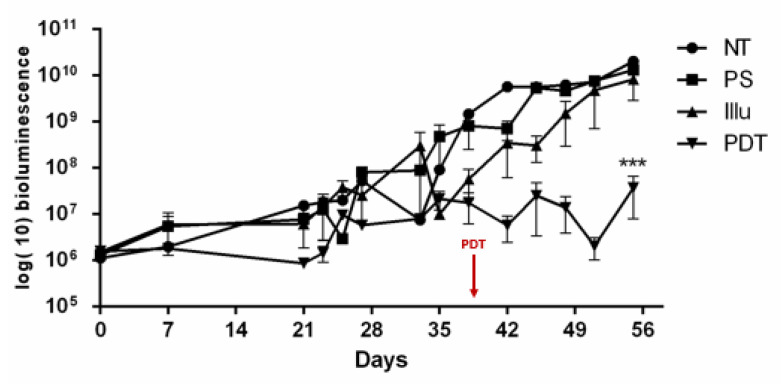
Tumor progression upon PDT for non-reconstituted SCID mouse model of intraperitoneal ovarian cancer: mice were subjected to a fractionized illumination protocol: 1 min illumination followed by 2 min of pause (10 mW/cm^2^, 27 J/cm^2^) with NT: non treated, PS: photosensitizer only, Illu: illumination only, PDT: illumination in the presence of PS, arrow: day of illumination. PDT efficacy was examined over 56 days and gauged by bioluminescence quantification. Results are presented as means of three independent experiments and expressed in log(10) of bioluminescence. Two-way ANOVA statistical test was performed, all quoted *p*-values are two-sided, with *p* ≤ 0.001 (***) being considered statistically significant. *n* = 3.

**Figure 6 ijms-24-11288-f006:**
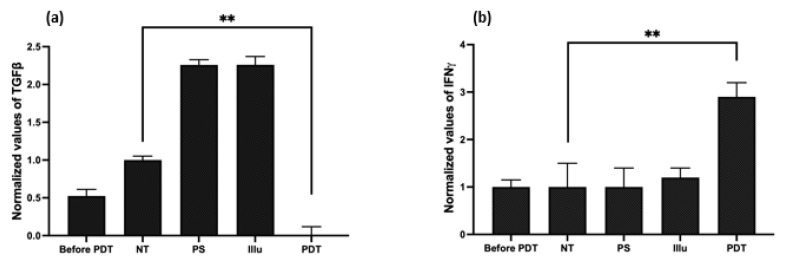
Evaluation of the cytokine release upon PDT in humanized non-reconstituted SCID mice model of intraperitoneal ovarian cancer: ELISA were performed on mice serum at 72 h after PDT (**a**) transforming growth factor (TGF) ß; (**b**) interferon (IFN) *γ* with NT: non treated, PS: photosensitizer only, Illu: illumination only, PDT: illumination in the presence of PS. Results are presented as means of three independent experiments expressed in % of the NT. One-way ANOVA statistical test was performed, all quoted *p*-values are two-sided, with *p* ≤ 0.001 (**) being considered statistically significant. *n* = 3.

**Figure 7 ijms-24-11288-f007:**
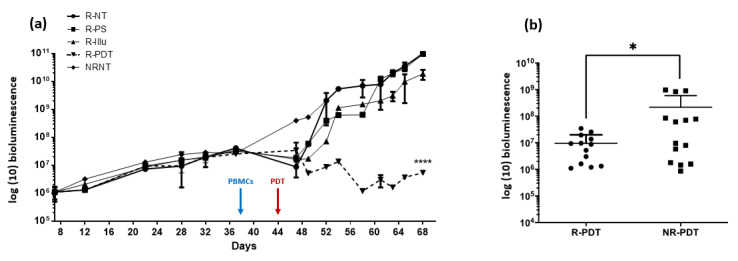
Evaluation of the PDT effect in PBMC reconstituted or not reconstituted humanized SCID mouse model of human intraperitoneal ovarian cancer: (**a**) bioluminescence in mice subjected to a fractionized illumination protocol: 1 min illumination followed by 2 min of pause repeated 45 times (10 mW/cm^2^, 27 J/cm^2^) with R-NT: reconstituted non treated, R-PS: reconstituted subjected to photosensitizer only, R-Illu: reconstituted subjected to illumination only, R-PDT: reconstituted subjected to illumination in the presence of PS, NRNT: non-reconstituted non treated, and arrow: day of illumination. Results are presented as means of three independent experiments. Two-way ANOVA statistical test was performed, all quoted *p*-values are two-sided, with *p* ≤ 0.0001 (****) being considered highly statistically significant. *n* = 3; (**b**) the overall tumor progression throughout the experiment was evaluated with R-PDT: reconstituted subjected to illumination in the presence of PS and NR-PDT: non-reconstituted subjected to illumination in the presence of PS. Results are presented as means of three independent experiments. Mann–Whitney U test was used to compare the two conditions with *p* ≤ 0.01 (*) being considered statistically significant.

**Figure 8 ijms-24-11288-f008:**
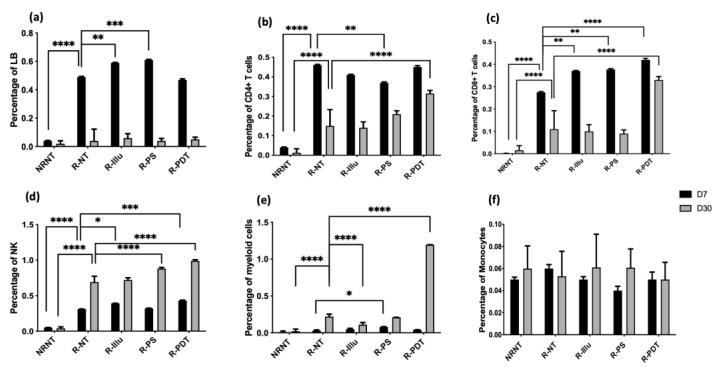
Evaluation of the immune response upon PDT in PBMC-reconstituted humanized SCID mouse model of peritoneal ovarian cancer: blood from not reconstituted (NR NT) or reconstituted mice subjected to illumination (R-PDT, R-Illu) or not (R-PS, R-NT) was examined and immune population was investigated 7 days (D7) and 30 days (D30) after PDT: (**a**) B cells (LB); (**b**) CD4+ T cells; (**c**) CD8+ T cells; (**d**) natural killers cells (NK); (**e**) myeloid cells; (**f**) monocytes with NRNT: non-reconstituted not treated, R-NT: reconstituted non treated, R-PS: reconstituted subjected to photosensitizer only, R-Illu: reconstituted subjected to illumination only, R-PDT: reconstituted subjected to illumination in the presence of photosensitizer. Two-way ANOVA statistical test was performed and comparisons with the R-NT conditions were done (at D7 and D30 separately), all quoted *p*-values are two-sided, with *p* ≤ 0.01 (*), *p* ≤ 0.01 (**), *p* ≤ 0.001 (***), and *p* ≤ 0.01 (****) being considered statistically significant for the first and (****) highly significant for the others.

**Figure 9 ijms-24-11288-f009:**
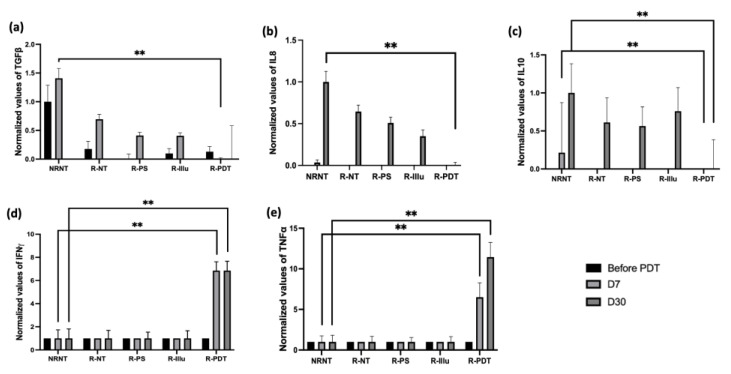
Evaluation of the cytokine release upon PDT in PBMC-reconstituted humanized SCID mouse model of peritoneal ovarian cancer: blood from not reconstituted (NR NT) or reconstituted mice subjected to illumination (R-PDT, R-Illu) or not (R-PS, R-NT) was examined for cytokine release 7 days (D7) and 30 days (D30) after PDT: (**a**) transforming growth factor ß (TGFß); (**b**) interleukine 8 (IL-8); (**c**) interleukin 10 (IL-10); (**d**) interferon *γ* (IFN*γ)*; (**e**): tumor necrosis factor α (TNFα) with NRNT: non–reconstituted non treated, R-NT: reconstituted non treated, R-PS: reconstituted subjected to photosensitizer only, R-illu: reconstituted subjected to illumination only, R-PDT: reconstituted subjected to illumination in the presence of photosensitizer. Results are presented as normalized values compared to NRNT. Two-way ANOVA statistical test was performed, all quoted *p*-values are two-sided, with *p* ≤ 0.001 (**) being considered statistically highly significant.

**Figure 10 ijms-24-11288-f010:**
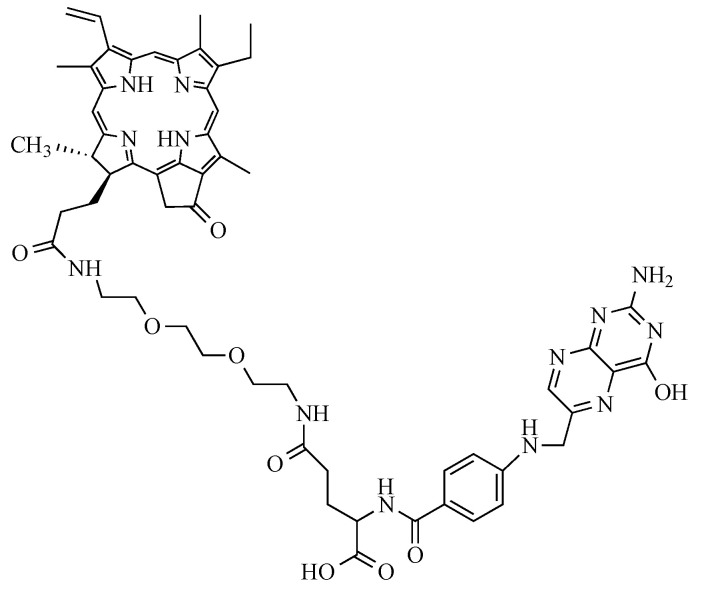
Structure of the Pyro-PEG-FA: folic acid conjugated to pyropheophorbide-a via a polyethylene glycol type spacer.

## Data Availability

The data sets analyzed are archived on a secure local compute server, available on request from the corresponding author.

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
