# Peer review of "Folate Receptor Targeted Photodynamic Therapy: A Novel Way to Stimulate Anti-Tumor Immune Response in Intraperitoneal Ovarian Cancer"

_ijms, 2023, doi:10.3390/ijms241411288_

Round 1

Reviewer 1 Report

This research Martha Baydoun & Nadira Delhem et al, addressed about N-{2-[2-(2-aminoethoxy)ethoxy]ethyl}folic acid}-4-carboxyphenylporphyrin (Porph-s-FA) PDT in the peritoneal disseminated ovarian cancer.

Major Points

#1

The authors should not use just PDT, but at least Porph-s-FA-PDT??? (or please NAME a particular PDT), in the article.

#2

The effect of this therapy is too small (p = 0.00469).

#3

The hydrophobic structure of 4-carboxyphenylporphyrin have no advantage in the pharmacokinetics and pharmacodynamics in the body.

#4

You should show the structure of the Porph-s-FA-PDT and the synthesis in method.

#5

Serum and ascites stability of Porph-s-FA-PDT should be shown

#6

The Figure 1 is not necessary, it should be moved to supplementary.

Moreover, the difference of FR receptor expression between OVCAR3 Luc and OVCAR3.

#7

Although the BIOD showed in Photodiagnosis Photodyn Ther. 2016 Mar;13:130-138. doi: 10.1016/j.pdpdt.2015.07.005., Please characterize the biodistribution in your model.

#8

In Figure4, only the Ovaries and tumors have been showed. Liver, kidney, intestine, stomach, pancreas, gall bladder etc,,  should be presented,

#9

The efficacy in vitro in Figure 2 is very good, however, the efficacy in vivo in Figure6 is very limited. How different the efficacy is!!  Please discuss and clarify the gap.

#10

In Figure 7, TGFb suppressed in the model. What is the mechanism???  Also Please show the other inflammatory cytokines.

#11

In Figure10, Please show the other inflammatory cytokines.

#12

Why the model mouse with reconstitution of immune system is used in this study??

You should test in syngeneic mouse tumor model.

#13

The authors described ROS is the mechanism pf this PDT.

The ROS generation data should be presented and evaluated.

Author Response

Dear Reviewer, you will find all of our responses on the linked file.

Reviewer 2 Report

The manuscript is very interesting and focuses on important clinical problem i.e. ovarian cancer - the most fatal gynaecological malignancy. 

Several minor [points, however, need to be elucidated:

1.      Statistical section should be supplemented with information concerning use statistical methodology – types of used tests are  described in the footnotes of tables – in my opinion however needs to be painted out in the main text

2.      Potential influence of samples’ size and number of laboratory determinations should be discussed if the number is sufficient (not to low) to obtain reliable results.

Author Response

(The authors gave the same response as above.)

Round 2

Reviewer 1 Report

This research Martha Baydoun & Nadira Delhem et al, addressed about N-{2-[2-(2-aminoethoxy)ethoxy]ethyl}folic acid}-4-carboxyphenylporphyrin (Porph-s-FA) PDT in the peritoneal disseminated ovarian cancer.

#1

Revised

#2

The authors did not precisely respond our questions. The effect is too small (p = 0.0469), which could not work in human clinical trials. You reply the immune-response of your therapy, in this case, you should show the total effect including the immune-reaction. Your reply is out of focus.

#3

If you agree the limited pharmacokinetics and pharmacodynamics, you should discuss with the limitation and describe the improvement.

#5

Do the experiment with ascites and serum.

#6

Please do the experiment to evaluate the expression between wild OVCAR3 and OVCAR luc.

#7

You should evaluate the organ damage after the therapy (ie. Kidney liver, ovary, colon). I think some damage exist.

#9

You should discuss the small effect in vivo, this agent cannot use in clinic.

#10

Do the experiment to show the mechanism!!

#11

Please show and indicate the other cytokines in the Figure. Do not data not shown!!

#13

Please do the evaluation of ROS in vitro and in vivo (ex vivo). There are many experimentevaluation ways to evaluate the ROS.

Author Response

Please find attached our reply to your comments.

Regards,

Round 3

Reviewer 1 Report

Revised.